# Role of Stress-Related Dopamine Transmission in Building and Maintaining a Protective Cognitive Reserve

**DOI:** 10.3390/brainsci12020246

**Published:** 2022-02-11

**Authors:** Simona Cabib, Claudio Latagliata, Cristina Orsini

**Affiliations:** 1Department of Experimental Neurosciences, IRCCS Fondazione Santa Lucia, 00179 Roma, Italy; claudio.latagliata@gmail.com; 2Department of Psychology, Sapienza University of Rome, 00185 Roma, Italy; cristina.orsini@uniroma1.it

**Keywords:** avoidance, cognitive reserve, connectivity, controllability, coping, dopamine, large-scale brain networks

## Abstract

This short review presents the hypothesis that stress-dependent dopamine (DA) transmission contributes to developing and maintaining the brain network supporting a cognitive reserve. Research has shown that people with a greater cognitive reserve are better able to avoid symptoms of degenerative brain changes. The paper will review evidence that: (1) successful adaptation to stressors involves development and stabilization of effective but flexible coping strategies; (2) this process requires dynamic reorganization of functional networks in the adult brain; (3) DA transmission is amongst the principal mediators of this process; (4) age- and disease-dependent cognitive impairment is associated with dysfunctional connectivity both between and within these same networks as well as with reduced DA transmission.

## 1. Introduction

The concept of cognitive reserve (CR) was developed to explain why brain damage does not always predict the severity of cognitive dysfunctions. It was associated with a history of positive life experiences (higher education and work milestones, active and rewarding lifestyles) and IQ. Hence, these factors are considered proxies and used as a measure of CR in the absence of a validated marker. CR proxies are associated with a reduced risk of cognitive decline in normal aging and dementia [1,2,3]. Moreover, Alzheimer’s disease (AD) patients with CR proxies can show more advanced AD-type neuropathology but comparable clinical severity of cognitive impairment than those without CR proxies [2,4]. Studies have reported similar findings in different neurodegenerative diseases such as multiple sclerosis [5], Parkinson’s disease [6], and schizophrenia [7]. Therefore, there is an increasing interest in developing interventions capable of fostering CR in the treatment of cognitive decline. This goal requires identifying neural processes and mechanisms involved in developing and maintaining this phenotype [8].

It has been proposed that CR involves efficient and flexible utilization of different brain networks. An authoritative CR definition posits that proxies “…result in individual differences in the flexibility and adaptability of brain networks that may allow some people to cope better than others with age- or dementia-related brain changes” [1]. This definition could well refer to the outcome of a history of successful coping with acute stressful experiences. Indeed, stressors are rarely traumatic; they are primarily changes in lifestyles that require the adaptation of behavioral, affective, and cognitive strategies [9]. Thus, marriage and personal achievement score 50 and 25/100, respectively, on a standard stress severity scale [10,11]. Moreover, human and animal studies have demonstrated that whereas traumatic, repeated, or chronic uncontrollable/unavoidable stress experiences foster dysfunctional adaptation, successful coping experiences make adults resilient to further stress. Being stress-resilient does not mean being insensitive to stressful events but capable of successfully adapting to the specific challenge encountered [12,13,14,15,16,17].

Results from animal or human studies also support the prominent role of dopamine (DA) transmission in developing and stabilizing successful coping strategies [12,18,19,20]. Conversely, reduced DA transmission has been associated with depressive symptoms, including anhedonia and apathy, which characterize both AD and frontotemporal dementia [21,22,23,24,25]. Moreover, data collected in AD patients and animal models of AD support impaired DA transmission in cognitive impairment, and age-related decline in striatal DA has been associated with the cognitive decline seen in normal aging [26,27,28,29]. Finally, evidence from genetic AD models indicates reduced DA availability in the limbic systems that largely precedes the development of the classical AD profile [30].

This brief review proposes a role of DA transmission associated with successful stress coping in fostering the brain network needed to support CR. Therefore, it will be focused on the dynamic of coping with novel stress, the responses and plasticity of brain circuits engaged by successful coping with stressful experiences, and the ability of DA to drive, guide, and control the reorganization of these circuits. Moreover, the evidence discussed will mostly come from research in animal models because of the difficulties of measuring changes in DA neurotransmission in human subjects and the advantage of testing causal influences of brain DA transmission on behavioral phenotypes offered by animal models.

## 2. Learning to Cope with Stress

Organisms behaviorally deal with stressful events through coping strategies that terminate the physiological defensive responses needed but are expensive and dangerous in the long run (allostatic overload) [31]. Thus, most coping strategies aim to remove or avoid the source of stress, and they are successful when stressors are susceptible to the organisms’ actions (removable, escapable/avoidable, controllable). Nonetheless, some stressors cannot be removed or escaped, such as losing a loved one. In these cases, coping strategies aim to control the emotional arousal responsible for maintaining physiological stress responses [9,32]. 

Studies in animal models investigate controlling stress severity and duration and the interaction between stressors and individuals. Results obtained by these studies indicate that coping strategies are developed through trial and error and stabilized through learning-related processes. Virtually all the aversively motivated learning tasks used in animal research are stressful situations that can be escaped, avoided, or controlled by specific coping strategies. Rodents learn to escape from a water maze by swimming toward a hidden platform that can be found in specific locations and learn to avoid a shock by performing specific actions when a conditioned stimulus (CS) predicts its arrival. These are strategies rather than reactions to aversive stimuli. Indeed, finding a hidden platform may require the use of complex cognitive abilities [33], while learning to avoid a shock predicted by a CS requires inhibition of the freezing response, an evolutionary preserved defensive reaction toward potential threats, as well as control over the impulse to escape before CS presentation [34,35,36]. 

Coping strategies must be flexible to adapt to new experiences. Thus, freezing is usually the immediate response to an unpredicted change in the environment and allows the organism to evaluate impending threats while reducing the risks of being detected [37]. Escape is the subsequent reaction because it requires the appraisal of actual risk and inhibition of the freezing response [17]. However, organisms need to inhibit both of these reactive coping responses to develop proactive strategies that are adaptive in the specific context [38]. Finally, findings from animal studies indicate that adapting to a novel stressor requires extinction processes that involve learning to inhibit the expression of previously acquired successful strategies appraised as ineffective in the new context [39]. 

The experience of successful coping fosters the learning of a specific response and protects the organism from dysfunctional outcomes of subsequent encounters with pathogenic stressors. Rats that learn to temporarily control a shock experience by an instrumental escape response (wheel turning) do not show the behavioral and neurochemical effects of a social defeat experience occurring seven days later in a different context [16]. These findings were obtained with the so-called ‘triadic’ protocol developed by S.F. Maier [16,40]. In this protocol, pairs of rats are exposed to tail-shocks delivered at random intervals, while control rats are exposed to the apparatus without receiving shocks. Only one rat can temporarily block the shock delivery (exposed to escapable stress: ES) for both members of a shocked couple. Thus, the yoked rats (exposed to inescapable stress: IS) share with the ES rats the experience of the physical aspects of the stressor (intensity, duration, temporal distributions) but not the experience of successfully controlling it [16,40]. Results obtained with this protocol demonstrated the expression of the behavioral indices of a severe anxiety-depression syndrome by IS but not by ES rats. Moreover, whereas ES rats readily and persistently extinguished a newly acquired conditioned freezing when the CS was disassociated from the aversive experience, the IS rats acquired conditioned freezing as an inflexible and relapsing response. Finally, implementing a triadic-like protocol in human experiments (using a loud noise as a stressor) yielded similar findings [40]. 

The reviewed evidence supports the conclusion that the experience of successful coping with stressful events fosters the ability to develop flexible strategies and protect organisms against the risk of developing phenotypes associated with anxiety and depression through subsequent interactions with uncontrollable/unavoidable stressors.

## 3. The Neurocircuitry of Stress Coping

Findings from studies in animal models indicate that brain responses to a novel stressor allow the organism to stabilize the coping strategy that is most effective for that situation. Hormonal and neurochemical stress responses engage different brain networks by sequentially activating and inhibiting the different loops connecting prefrontal cortices with limbic and striatal targets. The outcomes of the attempts to cope with the stressful experience determine the dynamic of this process [15,41]. 

The amygdala orchestrates long-term learning stabilization (memory consolidation) under emotional arousal, by modulating the mnemonic activity and synaptic plasticity in several brain regions [42]. Noradrenergic stimulation of the amygdala, typically fostered by arousing experiences, enhances the consolidation of both striatum-dependent and hippocampus-dependent memory. Moreover, noradrenergic activation in the basolateral amygdala (BLA) is required for stress hormones (glucocorticoids) to influence memory processing dependent on prelimbic cortex (PL) interactions with the anterior insular cortex (aIC) and dorsal hippocampus [43]. On the other hand, partially overlapping competitive circuits allow for flexible adaptation of acquired responses to the ongoing experience. A circuit connecting the infralimbic cortex (IL) with a GABAergic nucleus of the amygdala supports the extinction of a consolidated freezing response when the CS ceases to predict the aversive experience, a process moderated by the IC [44,45]. Moreover, a circuit connecting the PL with the ventral striatum through the BLA allows for escape/avoidance responses to overcome freezing when needed [36,38]. 

As discussed, learning to successfully cope with adverse experiences fosters resilience to subsequent stress experiences. Thus, rats who have learned to stop shock delivery temporarily (ES rats) are protected against the dysfunctional sequalae fostered by the stressor and by subsequent experiences of uncontrollable stress. A PL-centered circuit supports these protective effects [16,46]. Indeed, a PL connection with the dorsal raphe nucleus (DRN) was shown to inhibit serotonin (5-HT) release from local neurons and prevent the dysfunctional outcomes of the stressful experience in ES. Moreover, a PL connection with the dorsomedial striatum (DMS) is responsible for the appraisal of the situation as ‘controllable’, which fosters structural plasticity (increase in dendritic spines) in the PL. This neuroplastic event, in turn, is required for the PL-DMS circuit to be activated in response to subsequent uncontrollable stressors [47]. Finally, stressors used to model psychiatric disturbances in experimental animals foster a decrease rather than an increase in structural connectivity within the PL [48,49,50]. 

Human neuroimaging data collected in recent years indicate that exposure to an experimental stressor fosters a reallocation of resources to a large-scale neurocognitive network known as the salience network (SN). Large-scale networks consist of self-organized co-activation of brain areas. They are identified through resting-state functional magnetic resonance imaging by correlating fluctuations in the blood oxygen signal in different brain areas. According to the results of the meta-analyses of functional neuroimaging data, SN nodes respond consistently to different salient stimuli, including aversive material, conditioned stimuli, and pain. Thus, the network is proposed to play a crucial role in identifying the most relevant internal and external stimuli to guide behavior appropriately [51,52,53]. Although the brain areas included in a specific network can differ slightly, due to the rapid increase in data that characterizes this area of research, major nodes of the SN network are the dorsal anterior Cingulate Cortex (daCC), the aIC, the striatum, the amygdala, and the DAergic brainstem nuclei [53,54,55]. Together, these structures complete a discrete cortico-striatal-thalamic-cortical loop evident at functional and structural levels [55,56]. Finally, substantial evidence indicates that human aCC and rodents PL play the same role in supporting threat responses and connect with the same limbic/striatal areas [57]. Therefore, the SN encompasses the circuits that mediate the development and stabilization of coping strategies in animal models.

SN is only one of the identified large-scale networks. Still, it appears to dynamically control the communication between the default-mode (DMN) and the central-executive (CEN) networks in both young and elderly healthy subjects [58]. DMN network is more activated during internally directed cognitive activities, such as self-monitoring and social functions. In contrast, CEN seems to be more activated by externally controlled higher-order cognitive functions, such as attention, working memory, and decision-making. Evidence supports the hypothesis that functional coupling between the large-scale brain networks is essential for successful cognition and coping across health and disease [54,58,59,60,61]. Moreover, a pivotal role for the activity of DMN and SN in individual coping styles has been reported [62]. Most importantly, for the topic of the present review, higher connectivity between major networks in the brain is associated with an effective CR in Alzheimer’s disease and very high cognitive performance in healthy aging [63,64,65,66]. Finally, although most of these reports are based on functional connectivity data, there is evidence of structural effects of CR proxies [3,65,67].

Together, these considerations support the conclusion that coping with acute stress is mediated by a brain network centered around PL and IC that engages the amygdala, the striatum, and DA transmission. Moreover, stabilization of flexible adaptive coping strategies requires neuroplasticity within specific nodes of these circuits. On the other hand, data from human studies indicate that SN, a large-scale network including daCC, aIC, the amygdala, the striatum, and the brain DA systems, is engaged by stress experiences and coping. Finally, functional and structural plasticity within and between the SN and other primary brain networks exert protection against cognitive deterioration fostered by aging ad disease.

## 4. Dopamine

Neural responses to environmental changes enable organisms to rapidly detect threats, respond adequately, restore homeostasis through successful coping, and better prepare for future challenges. Stress-sensitive hormones and neurotransmitters are dynamically activated and inhibited to modulate the neural excitability of different brain circuits. This complex response develops according to a highly preserved pattern across species and is sensitive to the outcomes of the ongoing interaction between the organism and the stressor. There are several informative reviews on stress hormones, norepinephrine (NE), 5-HT, and their effects on brain functioning and plasticity [20,68,69,70,71,72,73]. Therefore, the present review focuses on DA transmission.

The DA circuit involved in successful coping with acute stress is the mesocorticolimbic system made up by ventral tegmental area (VTA) DA neurons projecting to the PL, the amygdala, the ventral striatum (NAc), and the hippocampus. A recent review defined the role of the mesocorticolimbic circuitry: “…as a hub linking circuits involved in the emotional-motivational appraisal of salient information with networks underlying executive functioning” [20]. Indeed, the mesocorticolimbic circuit includes core regions of the SN, and it is in the position to engage the dorsal striatum through a dynamic ‘spiraling’ connection with the substantia nigra [74], the major source of striatal DA (Figure 1). Data collected in animal models point to a rapid but temporary increase in extracellular DA in PL, amygdala, and NAc in response to novel stress [19,75,76,77]. The NAc DA response to novel stressors is dependent on increased NE transmission in the PL and is constrained by DA transmission in the same brain area [19,78,79,80,81]. DA transmission in the NAc supports active coping by stimulating DA receptors of the D2 type [9,19,82] and is dependent on the expectancy of successful active coping. Indeed, during the experience of a novel stressful situation as inescapable and uncontrollable, DA release in the NAc is progressively inhibited by a sharp reduction in NE and a parallel increase in DA in the PL [19]. 

One of the hypothesized mechanisms by which brain DA acts is in mediating the dynamic balance between processes of “flexible updating and cognitive stabilization” [83]. As discussed, a PL-DMS circuit is engaged by the appraisal of stress controllability. Inactivation of each brain area does not influence either the expression or the acquisition of the specific coping response used to control the stressor. Instead, connectivity between PL and DMS is required for the protective effects of the previous experience of control against dysfunctional outcomes of subsequent inescapable-uncontrollable stressors [16]. A PL-DMS connection is also involved in the acquisition of positively valued goal-directed instrumental actions, which flexibly adapt to changes in the value of the goal. Instead, connectivity between motor cortices and the dorsolateral striatum (DLS) is required to acquire a habitual instrumental response elicited by conditioned stimuli regardless of the goal value [84]. All instrumental responses are acquired in parallel by the two circuits, which then compete to control the expression of instrumental responses. DA transmission in DMS or DLS will determine whether the organism expresses the acquired goal-directed or habitual response [84]. It is thus possible that rats exposed to controllable stress engage the circuit responsible for habitual behavior to express a specific escape response but engage the circuit responsible for goal-directed behavior to adapt to novel stressful conditions. In line with this hypothesis, six weeks of voluntary wheel running, which protects rats against the adverse effects of uncontrollable stress as the acute experience of control does, were shown to significantly enhance DA release induced by an uncontrollable shock in the DMS but not in DLS [85]. 

An acute stress experience or the administration of anxiogenic compounds increases DA output in the PL/IL area of Roman high-avoidance (RHA/Verh) but not of Roman low-avoidance rats (RLA/Verh). RHA/Verh display robust and long-lasting active coping with novel stressors, whereas RLA/Verh rats show frequent freezing that interferes with acquiring the active avoidance response [86]. DA transmission in the PL does not mediate the acquisition of a positively reinforced instrumental response, but it is necessary for learning about action–outcome associations [50]. Thus, DA transmission within the PL could guide the choice of the most effective coping strategies in novel stressful situations. In line with this hypothesis, a specific PL-DMS pathway is involved in decision-making about motivationally conflicting options [87]. 

Rats with experience of control over a stressor show increased dendritic spines within PL [47]. As discussed, experimental stressful experiences used to reproduce dysfunctional phenotypes in animal models foster increased structural neuroplasticity in striatal/limbic targets but reduced plasticity in PL [48,49,50,88] and the PL-DMS circuit is disrupted by the experience of these stressors [89]. Structural plasticity in the adult brain relies on the experience-dependent proliferation and pruning of dendritic spines within specific networks, leading to reorganization of connectivity between neuronal populations. DA is required for glutamate-induced spinogenesis and exerts dichotomous effects on neurons expressing the D1 or D2 receptor subtypes so that they store memories of reward and reward omission, respectively, through cell-type-specific spine enlargement [90,91]. Finally, reduced DA availability due to degeneration of DA cells interferes with spinogenesis while increasing pruning [92]. 

Moreover, there is strong evidence that DA transmission modulates SN connectivity [93,94,95]. DA modulates aCC activity during executive tasks and the synaptic availability of DA may be directly related to efficient IC function. On the other hand, a high correlation between the binding of the D2/D3 ligand [18F]-fallypride and grey matter density has been observed in the aCC, IC, and midbrain, suggesting that reduced grey matter observed in the SN of schizophrenics is directly associated with DA deficits [95]. Moreover, DA synthesis capacity has been recently associated with greater SN connectivity, particularly in brain regions that act as information-processing hubs [94] and there is strong evidence for VTA functional connectivity with the posterior cingulate/precuneus, a central hub node of the DMN [96]. On the other hand, progressive depletion of striatal DA is the hallmark of Parkinson’s disease (PD). Around 30% of PD patients present cognitive deficits long before the appearance of classic motor symptoms and evidence from clinical and preclinical studies on cognitive deficits in PD points to a role of reduced striatal DA availability [6,97]. There is strong evidence that reduced DA availability in the ventral striatum (NAc) precedes typical motor deficits in PD and several studies found a reduced connectivity between the aCC and the ventral striatum [98]. Finally, PD patients with mild cognitive impairment, who are susceptible to developing dementia, are characterized by severe DA depletion in the associative striatum and reduced D2 receptor availability in the IC [99]. These findings seem coherent with the dynamic relationship between PL and NAc DA observed in stressed animals and bridge dysfunctional connectivity within SN and between large brain networks with cognitive impairment and reduced DA transmission.

Together, the discussed evidence points to a significant role of DA transmission in cortical, limbic, and striatal nodes in the immediate and long-term adaptive effects of successful coping with stress challenges. Moreover, the reviewed evidence suggests that DA exerts this role by modulating both functional and structural brain connectivity within the SN and supports a strong influence of this modulation on cognitive capacity.

## 5. Conclusions

This brief review discussed evidence of an overlap between brain circuitry supporting the development and stabilization of adaptive stress coping strategies and cognitive functioning in aging and disease. Moreover, we pointed to a significant moderating role of DA transmission in specific nodes of the SN in circuital functioning and plasticity in the adult brain. Based on these considerations, it is reasonable to hypothesize that experiences of successful coping with life events, through DA transmission-induced modulation of brain functional and structural connectivity, contribute to establishing a rich CR. Instead, reduced brain DA availability, due to genetic predisposition or to a history of uncontrollable/unavoidable experiences, can undermine the development and maintenance of CR, leading to more severe and rapid cognitive deterioration in response to neurodegeneration. Finally, the proposed hypothesis opens new opportunities to translational research by overcoming the problems of reproducing classic CR proxies in animal models. 

## Figures and Tables

**Figure 1 brainsci-12-00246-f001:**
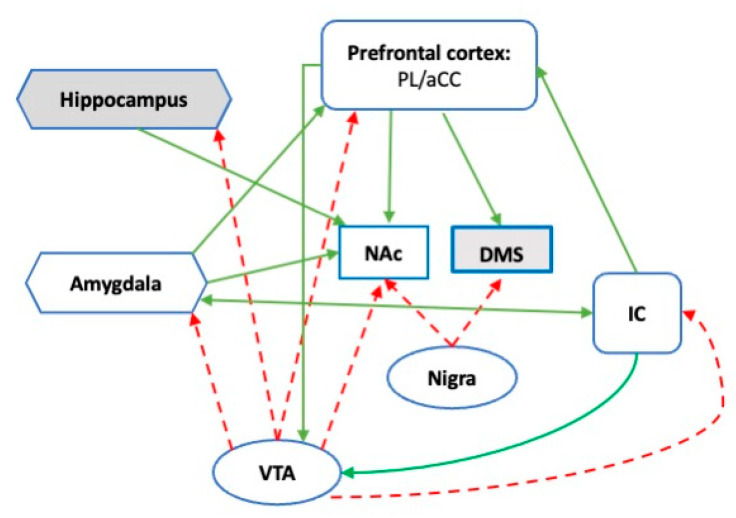
Schematic representation of the mesocorticolimbic DA system associated with stress coping. Dotted lines indicate DA afferences; solid lines indicate functional connectivity; white-filled areas are part of the SN in humans and rodents [53]. PL prelimbic cortex; aCC: Anterior cingulate cortex; IC: Insula cortex; NAc: Nucleus accumbens/ventral striatum; DMS: Dorsomedial striatum; Nigra: Substantia nigra pars compacta; VTA: Ventral tegmental area.

## Data Availability

Not applicable.

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
