# Peer review of "Role of Stress-Related Dopamine Transmission in Building and Maintaining a Protective Cognitive Reserve"

_brainsci, 2022, doi:10.3390/brainsci12020246_

Round 1

Reviewer 1 Report

The review by Simona Cabib, Claudio Latagliata and Cristina Orsini on the role of stress-related dopamine transmission in building and maintaining a protective cognitive reserve is an interesting review that presents the hypothesis that stress-dependent dopamine transmission contributes to developing and maintaining the brain network supporting a cognitive reserve.

The review is clear and comprehensive. As no similar review has been published recently, this work appears to be needed and relevant to the field and provides a good summary of the current state of knowledge. The references are adequate and current (about 60% of them were published within the last 5 years).

The review is divided into several chapters, which are: (1) ‘Introduction’ - the authors explain the concept of cognitive reserve and emphasize its importance for preventing / treating cognitive decline, and point out that dopamine appears to play an important role in creating and maintaining this reserve; (2) ‘Learning to cope with stress’ - the authors cite the results of various studies in animal stress models, emphasizing that the experience of success in coping with stressful events fosters the ability to develop flexible strategies and protect the organism against the risk of developing phenotypes associated with anxiety and depression; (3) ‘The neurocircuitry of stress coping’ - where, as the title suggests, the authors describe in detail which brain structures and signaling pathways may be responsible for coping with stress; (4) ‘Dopamine’ - which shows evidence of the significant moderating role of dopamine transmission in cortical, limbic, and striatal nodes in circuit functioning and plasticity in the adult brain.  

Some minor remarks:

  1. If the aim of the work is to emphasize the role of dopamine in building and maintaining the cognitive reserve, keywords such as ‘dopamine’ or ‘dopamine transmission’ and ‘cognitive reserve’ should also be included.
  2. It seems that paragraphs 4 and 5 in Chapter 2 should be combined (the current paragraph 5 is a simple continuation of some results description that was started in paragraph 4).
  3. Caption of Figure 1 – the list of abbreviations used in the figure is missing (IC, NAc, DMS, DLS, PL/ACC, VTA). I am aware that they are listed in the main text, but placing them below the figure will make it easier to read.
  4. The additional figure showing the neurocircuitry associated with stress coping is missing in the article (Chapter 3). Such a figure would be extremely valuable to understand these complex relationships between different areas of the brain that are involved in these processes.
  5. Word repetition – the authors should pay attention to quite numerous repetitions of some expressions, such as ‘indeed’ used more than 10 times, which in some sentences could easily be replaced with synonymus. Such stylistic corrections will certainly have a positive impact on the reception of the paper.

Author Response

We wish to thank the reviewer for the careful reading of the paper and useful suggestions.

We apported the following corrections

1. If the aim of the work is to emphasize the role of dopamine in building and maintaining the cognitive reserve, keywords such as ‘dopamine’ or ‘dopamine transmission’ and ‘cognitive reserve’ should also be included.

In the new version ‘dopamine’ and ‘cognitive reserve’ were added to the list of passwords.

  1. It seems that paragraphs 4 and 5 in Chapter 2 should be combined (the current paragraph 5 is a simple continuation of some results description that was started in paragraph 4).

In the new version paragraphs 4 and 5 were unified

  1. Caption of Figure 1 – the list of abbreviations used in the figure is missing (IC, NAc, DMS, DLS, PL/ACC, VTA). I am aware that they are listed in the main text but placing them below the figure will make it easier to read. The additional figure showing the neurocircuitry associated with stress coping is missing in the article (Chapter 3). Such a figure would be extremely valuable to understand these complex relationships between different areas of the brain that are involved in these processes.

Both reviewers pointed to the limits of the caption of Figure 1. This figure is the schematic representation of the circuit involved in stress coping. The new caption should clarify this point.

  1. Word repetition – the authors should pay attention to quite numerous repetitions of some expressions, such as ‘indeed’ used more than 10 times, which in some sentences could easily be replaced with synonymus. Such stylistic corrections will certainly have a positive impact on the reception of the paper.

The new version was thoroughly checked for repetitions

As reviewer 2 asked to elaborate more on the relationships between stressor coping and Parkinson’s disease since Parkinson’s disease is caused by dopamine deficiency.

We added a discussion of Parkinson data to the new version of the MS to meet this request. To host this addendum, we changed the organization and some parts of the Dopamine chapter.

We apported the following corrections

  1. If the aim of the work is to emphasize the role of dopamine in building and maintaining the cognitive reserve, keywords such as ‘dopamine’ or ‘dopamine transmission’ and ‘cognitive reserve’ should also be included.

In the new version ‘dopamine’ and ‘cognitive reserve’ were added to the list of passwords.

2. It seems that paragraphs 4 and 5 in Chapter 2 should be combined (the current paragraph 5 is a simple continuation of some results description that was started in paragraph 4).

In the new version paragraphs 4 and 5 were unified

3. Caption of Figure 1 – the list of abbreviations used in the figure is missing (IC, NAc, DMS, DLS, PL/ACC, VTA). I am aware that they are listed in the main text but placing them below the figure will make it easier to read. The additional figure showing the neurocircuitry associated with stress coping is missing in the article (Chapter 3). Such a figure would be extremely valuable to understand these complex relationships between different areas of the brain that are involved in these processes.

Both reviewers pointed to the limits of the caption of Figure 1. This figure is the schematic representation of the circuit involved in stress coping. The new caption should clarify this point.

4. Word repetition – the authors should pay attention to quite numerous repetitions of some expressions, such as ‘indeed’ used more than 10 times, which in some sentences could easily be replaced with synonymus. Such stylistic corrections will certainly have a positive impact on the reception of the paper.

The new version was thoroughly checked for repetitions

Reviewer 2 Report

The manuscript is interesting to the readers. The review is quite comprehensive, but a few things need to be addressed.

  1. More explanations for Fig. 1 are needed. Full names are needed for the abbreviations.
  2. It is important to elaborate more on the relationships between stressor coping and Parkinson’s disease since Parkinson’s disease is caused by dopamine deficiency.

Author Response

We wish to thank the reviewer for the careful reading of the paper and useful suggestions.

We apported the following corrections

  1. More explanations for Fig. 1 are needed. Full names are needed for the abbreviations.

This request was met in the new version, we also changed the figure to render it more informative

  1. It is important to elaborate more on the relationships between stressor coping and Parkinson’s disease since Parkinson’s disease is caused by dopamine deficiency.

We added a discussion of Parkinson data to the new version of the MS to meet this request. To host this addendum we changed the organization and some parts of the section: Dopamine